# Tramtrack acts during late pupal development to direct ant caste identity

**Karl M. Glastad**[1,2], **Linyang Ju**[2,3], **Shelley L. Berger**[1,2,3]*

**1** Department of Cell and Developmental Biology, Department of Genetics, Perelman School of Medicine, University of Pennsylvania, Philadelphia, Pennsylvania United States of America, **2** Epigenetics Institute; Perelman School of Medicine, University of Pennsylvania, Philadelphia, Pennsylvania United States of America, **3** Department of Biology, School of Arts and Sciences, University of Pennsylvania, Philadelphia, Pennsylvania United States of America

* bergers@pennmedicine.upenn.edu

**Data Availability Statement:** All sequencing data related to this project has been deposited in the NCBI SRA BioProject: PRJNA726183.

**Funding:** K.M.G. was supported by a NIH training grant (F32 GM120933, National institute of general

## Abstract

A key question in the rising field of neuroepigenetics is how behavioral plasticity is established and maintained in the developing CNS of multicellular organisms. Behavior is controlled through systemic changes in hormonal signaling, cell-specific regulation of gene expression, and changes in neuronal connections in the nervous system, however the link between these pathways is unclear. In the ant *Camponotus floridanus*, the epigenetic corepressor CoREST is a central player in experimentally-induced reprogramming of caste-specific behavior, from soldier (Major worker) to forager (Minor worker). Here, we show this pathway is engaged naturally on a large genomic scale during late pupal development targeting multiple genes differentially expressed between castes, and central to this mechanism is the protein tramtrack (ttk), a DNA binding partner of CoREST. Caste-specific differences in DNA binding of ttk co-binding with CoREST correlate with caste-biased gene expression both in the late pupal stage and immediately after eclosion. However, we find a unique set of exclusive Minor-bound genes that show ttk pre-binding in the late pupal stage preceding CoREST binding, followed by caste-specific gene repression on the first day of eclosion. In addition, we show that ttk binding correlates with neurogenic Notch signaling, and that specific ttk binding between castes is enriched for regulatory sites associated with hormonal function. Overall our findings elucidate a pathway of transcription factor binding leading to a repressive epigenetic axis that lies at the crux of development and hormonal signaling to define worker caste identity in *C. floridanus*.

## Author summary

The ability to dynamically adapt behavior in response to a variable environment is one of the hallmarks of complex life. Ants provide an excellent system within which to explore the molecular mechanisms governing such behavioral plasticity in the context of complex societies. We find that in the Florida Carpenter Ant *C. floridanus* two members of a known repressive complex, CoREST and ttk, are predictive of early-life gene expression differences between Major and Minor workers, which display vastly different behaviors.

medical sciences, https://www.nigms.nih.gov/).
This work was supported by funding from the
National Institute of Aging (R01 5R01AG055570,
National Institute of Aging, https://www.nia.nih.
gov/, to S.L.B.). The funders had no role in study
design, data collection and analysis, decision to
publish, or preparation of the manuscript.

**Competing interests:** The authors have declared
that no competing interests exist.

Furthermore, we find that ttk shows a signal of poising genes in the late pupal stage for
repression via CoREST upon eclosion, illustrating an order of operations in the establish-
ment of distinct behaviors that distinguish these two worker types. Regions bound by ttk
in Minor but not Major workers also show enrichment for sequence motifs associated
with hormonal regulation, implying involvement of ttk-mediated repression in attenuat-
ing hormonal signaling between castes. Given that differences in hormone signaling
between castes have been seen in multiple social insects our results provide evidence of a
chromatin bridge between hormones and behavior in a complex social organism.

## Introduction

Based upon environmental cues, members of a eusocial insect colony have the ability to differ-
entiate into morphologically and behaviorally-distinct sterile (worker) or reproductive (queen)
physiological castes [1]. Further, among many eusocial insect species, there exists a further
division of labor represented by the allocation of distinct colony roles among specific worker
groups [1–5]. Thus, despite sharing genotypes, members of a colony exhibit dramatic pheno-
typic and behavioral distinctions more akin to differences between species. Because of this
extreme phenotypic and behavioral plasticity in the context of highly related individuals, euso-
cial insects have emerged as models for interrogating complex social behavior, and for investi-
gating molecular mechanisms that program this remarkable plasticity [6, 7].

Epigeneic mechanisms provide a molecular basis for eusocial insect plasticity. Indeed, dis-
tinct patterns of chromatin modifications decorating both DNA and histone proteins correlate
with differential morphology and behavior among social insect castes [2, 8–15]. Central to the
establishment of differential chromatin modification are transcription factors (TFs) that
recruit co-factors that in turn modify chromatin or provide a scaffold for assembly of modifi-
ers; these gene regulatory assemblies serve as gene-proximal signal integrators that activate
transcription and then lead to longer-lasting changes in underlying chromatin for cell type
specification [16, 17].

Circulating hormones are an important class of signals that can generate large changes in
chromatin and cell state upon ligand binding [18]. Hormones play important roles in regulat-
ing development [19, 20] and behavior [21, 22], at least in part through manipulation of the
epigenome [23–26]. In eusocial insects the classic metamorphosis-associated hormones Juve-
nile Hormone (JH3) and ecdysone (20E) have emerged as important mediators of caste, in
governing developmental trajectory [27] and behavioral division of labor [28–33]. Juvenile
hormone in particular is associated with production of queen (reproductive caste) [30], and of
distinct worker castes in honeybees and ants [29, 31]. For example, in honeybees and ants
increased JH3 correlates with age-associated transition to foraging in workers [34], and thus
serves a key role in organizing colony behavioral division of labor. In spite of the central role
of hormones in caste specification, the transcriptional pathways governing hormone produc-
tion are not understood.

The Florida carpenter ant *Camponotus floridanus* has two distinct worker castes: Minor
and Major workers, which exhibit distinct behavioral repetoires. The smaller Minor workers
typically fill the role of the classical worker ant, performing most colony tasks such as foraging
for food and nursing of brood. In contrast, the larger Major workers defend the nest, serving
as soldiers and very rarely foraging [11]. Strikingly, this behavioral dichotomy is amenable to
experimental manipulation: Major workers can be reprogrammed to forage, using chromatin-
based manipulation via genetic and pharmacological tools [11], suggesting that behavioral

differentiation of worker caste remains malleable long after larval stage phenotypic specification [35].

Previously we demonstrated that programming of foraging behavior in *C. floridanus* Minor worker, and reprogramming of Major worker behavior toward Minor-like foraging is mediated by the neuronal co-repressor CoREST via regulation of caste-biased gene expression [2]. Notably, this reprogramming was most potent very early after eclosion from the pupal case, while Majors older than 10 days showed very low levels of re-programming. Here we demonstrate that the DNA binding factor tramtrack (ttk)[36], acting with CoREST, is implicated in controlling caste-biased gene regulation at the late pupal stage and immediately following eclosion. Strikingly, for a subset of largely Minor-biased genes, ttk acts first in the pupal stage to prime caste-specific genes for later repression and to serve as a key downstream regulator of hormonal signaling. Given high conservation in these pathways, our results may provide insight into general transcriptional mechanisms underlying the modulation of behavioral and developmental plasticity in animals.

## Results

### Transcriptome of the late pupal stage (LPS) shows strong signal of caste

Previously, we found, using RT-qPCR, that CoREST and several other genes related to caste division of labor are differentially expressed in the late pupal stage (hereafter: LPS) 0–1 days prior to eclosion [2]. In conjunction with the large number of gene expression differences seen at d0 which are greatly reduced by d5 [2], we hypothesized that the LPS-d0 ages represent a crucial time for behavioral specification in natural caste development, and that CoREST may act definitively at this stage compared to later adult timepoints. To globally assess the extent to which caste specification occurs prior to adulthood we performed RNA-sequencing (RNA-seq) on single brains of Major and Minor worker pupa (n = 5), focusing on the LPS, and compared these LPS data to RNA-seq data from our previous study of day 0 just following eclosion (d0) [2].

We found genes differentially expressed between Major and Minor in the LPS (730 DEGs, 300 Major-biased, 420 Minor biased; S1 Table) were strongly enriched for functional terms of neurodevelopment, behavior, and hormonal regulation (Fig 1A, 1B and S2 Table), consistent with the LPS serving as a formative time in *C. floridanus* social development. Interestingly, the top functional term enriched among LPS caste-biased genes was "R7 cell development" (Fig 1A) implicating photoreceptor development in pupal caste differentiation; however closer inspection of the genes contributing to this term revealed genes with many neurodevelopmental roles in fly (S1 Table), potentially reflecting annotation bias associated with this well-characterized neuronal model in fly.

In RNA-seq, CoREST and its binding partner tramtrack (ttk; transcription factor REST functional homolog) were both differentially expressed in the LPS, and showed higher expression in Minor compared to Major (Fig 1B and 1C). In contrast, the key JH-degrading enzymes encoded by Jhe and Jheh genes showed higher expression in Major (Fig 1B and 1C). In further support for the LPS being an impactful time in caste neurological development, we observed that multiple other genes related to caste division of labor in previous studies were differentially expressed, and these point to a consistent finding that Minor workers exhibit higher JH3 levels (as we showed previously [2]). Genes expressed higher in Minors include Kruppel homolog 1 (*Kr-h1*), a downstream effector of JH3 signaling [37]. In comparison, Majors showed higher expression of Ilp3, Impl2 (genes central to regulation of insulin signaling), and the foraging gene (*for*) (Fig 1B). Insulin signaling is strongly linked to eusocial insect division of labor; in particular insulin signaling is higher in reproductive castes of many eusocial ants

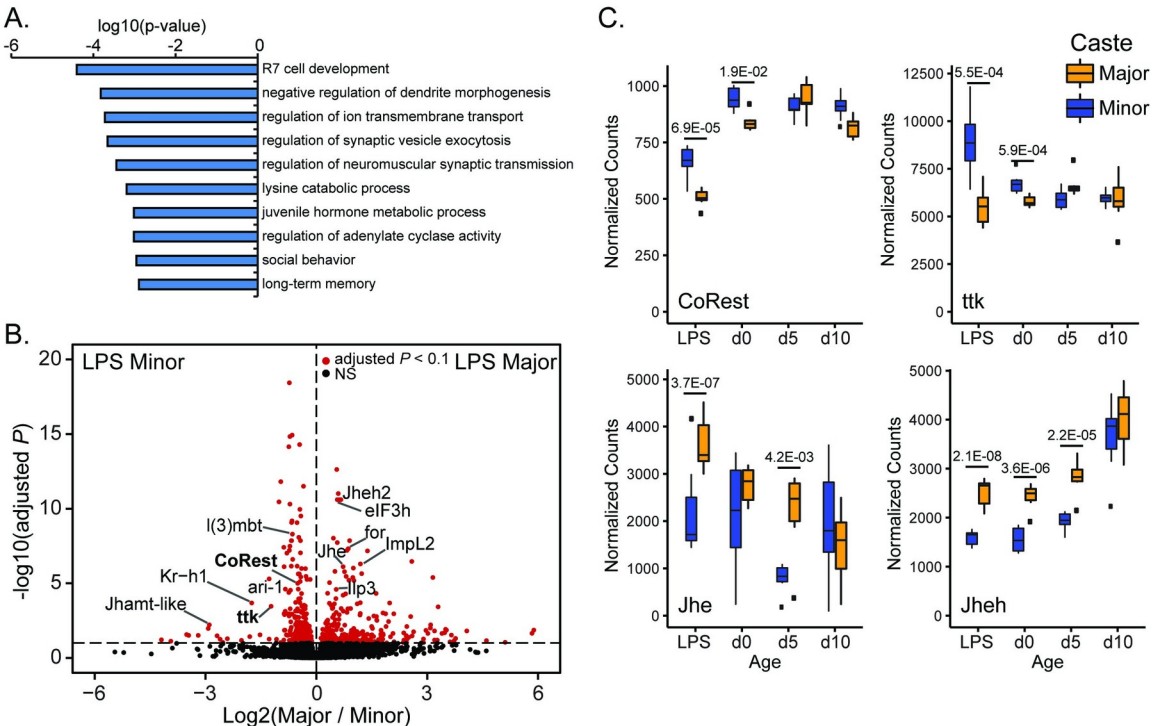

**Fig 1. Late pupal stage shows transcriptional differentiation of Major and Minor castes, and CoREST and its binding partner tramtrack show differential expression in the late pupal stage.** A) Top 10 GO categories (Biological Process) enriched among genes significantly differing (RNA-seq adjusted p-value < 0.1) between late pupal Major and Minor worker brains. B) Volcano plot of all DEGs identified between Major and Minor brains in the late pupal stage (from 7 biological replicates each), with genes related to hormonal signaling and CoREST labeled. Genes showing an adjusted p-value < 0.1 are highlighted in red. C) Gene-level plots of normalized counts for CoREST and binding partner ttk as well as two key JH-degrading genes plotted for late pupal stage (LPS), as well as d0, d5, and d10 post-eclosion. Data from d0, d5 and d10 reanalyzed from [2]. Major workers' values are given in orange and Minor workers' in blue.

[38] and underlies queen development [39, 40], and worker division of labor in honey bees [41]. Interestingly, the *for* gene in particular is implicated in regulating insulin signaling in the fly [42], and associated with differences in foraging behavior in social insects [43–45], and with insulin signaling in reproductive division of labor[38]. Thus, our data support LPS as a formative stage in behavioral divergence of Major and Minor workers, and principally correlating with differential hormonal signaling, where Minors show a signature of higher JH3 signaling and Majors show higher levels of insulin signaling.

## CoREST and tramtrack repress genes in a caste-specific manner

Because we observed CoREST and its binding partner ttk to be significantly higher in expression in Minor in both the LPS and d0 (Fig 1C), we evaluated their binding to the genome at these stages in Major and Minor workers. Importantly, in *D. melanogaster*, ttk acts as a functional homolog to the vertebrate gene REST, which is a DNA binding transcription factor recruiting CoREST to a subset of CoREST-repressed genes [36]. To test binding of ttk, we developed an antibody to *C. floridanus* ttk (validation shown in S1A Fig), and, in combination with our antibody targeting *C. floridanus* CoREST [2], performed CUT&RUN [46] on brains of Major and Minor from the LPS and d0 post-eclosion. Overall we found 7,620 CoREST peaks localizing to 6,664 genes, and 4,541 ttk peaks localizing to 3,983 genes (S1B Fig and S3 Table). Genes bound by either CoREST or ttk were enriched for a myriad of functional terms

related to transcriptional regulation, development, or neuronal functions (S4 Table). Notably, regardless of caste-bias, general peaks for both ttk and CoREST significantly intersected differentially expressed genes (DEGs) in both the LPS and d0 (S1B Fig, right), suggesting that binding of CoREST and ttk are preferentially localized to genes that significantly differ between caste.

Supporting predominantly repressive roles in both castes, regions significantly differing between Major and Minor workers in CoREST and ttk binding showed similarly strong negative continuous associations with gene expression differences between Major and Minor workers in the LPS (Fig 2A) as well as highly significant overlap between differential binding and differential expression (Fig 2B). We previously found one paralogue of the Juvenile hormone esterase (Jhe) gene to be highly expressed in Major and that CoREST repressed its expression to promote reprogramming to Minor-like foraging [2]; here we found CoREST showed strong Minor-biased binding at this locus during the LPS and at d0, and ttk showed similar Minor-bias (Fig 2A and 2C; boxplots: RNA-seq expression in LPS). We observed multiple other genes showing strong differential gene expression in the LPS that also featured a CoREST+ttk peak bias toward binding in the opposite caste, including *for* (Minor-repressed, Major-biased binding; Fig 2C) and the MSL-associated chromatin-linked adaptor protein *Clamp* (Major-repressed, Minor-biased binding; Fig 2C).

Consistent with its role as a DNA binding partner of CoREST [36], we found that most (90%) ttk peaks coincided with a CoREST peak, but almost half of CoREST peaks did not overlap with ttk peaks (Fig 2D and S3 Table), suggesting other DNA binding proteins act with CoREST at these latter sites. Peaks with both ttk and CoREST showed enrichment for G/C-rich motifs, with the top scoring *de novo* motif showing similarity to the known ttk binding motif in fly, suggesting conserved DNA binding specificity, while the top motif associated with CoREST peaks *lacking* ttk showed high similarity to the fly motif for the gene Drop (Dr; Fig 2D), a homeodomain TF known in fly to be involved in neuroectoderm patterning and neuronal and glial cell development [47]. Interestingly, genes showing high ttk binding (relative to genes possessing only CoREST) showed functional enrichment for multiple developmental terms, suggesting genes bound by both ttk and CoREST have neurodevelopmental roles (Fig 2E and S5 Table) more than genes bound only by CoREST (S6 Table).

In comparing Minor to Major we found 346 regions differentially bound by CoREST in the LPS (153 and 193 Major- and Minor-biased, respectively) and 233 at d0 (86 and 147 Major- and Minor-biased, respectively) localized to 330 and 214 genes respectively (S7 and S8 Tables). For ttk we found a similar number of differentially bound regions (DBRs), with 256 in LPS (125 and 131, Major- and Minor-biased respectively) and 314 at d0 (135 and 179 Major- and Minor-biased respectively). Thus, only a subset of CoREST or ttk peaks show caste differential binding, suggesting the core neurodevelopmental roles of CoREST and ttk are maintained in *C. floridanus*. Genes differentially bound by CoREST or ttk in either stage showed general enrichment for Gene Ontology terms related to neuronal function, development, or neurodevelopment when compared to genes bound but unchanging for the respective TF (S1C Fig), supporting the conclusion that these genes play important roles in the neurodevelopment of castes.

In *D. melanogaster* ttk acts downstream of Notch signaling [48–50], and in honey bees Notch functions in reproductive division of labor to restrain worker reproduction [51]. Notch signaling has an important role in neuronal differentiation, gliogenesis, and adult brain function [48, 50, 52, 53], thus differences in Notch signaling during the formative LPS and d0 stages may be associated with differences in brain size and function between castes. Thus, we examined whether Minor workers possess a signature of enhanced Notch signaling, using flybase-annotated genetic interactions between Notch and downstream genes [54]. Notably, we found

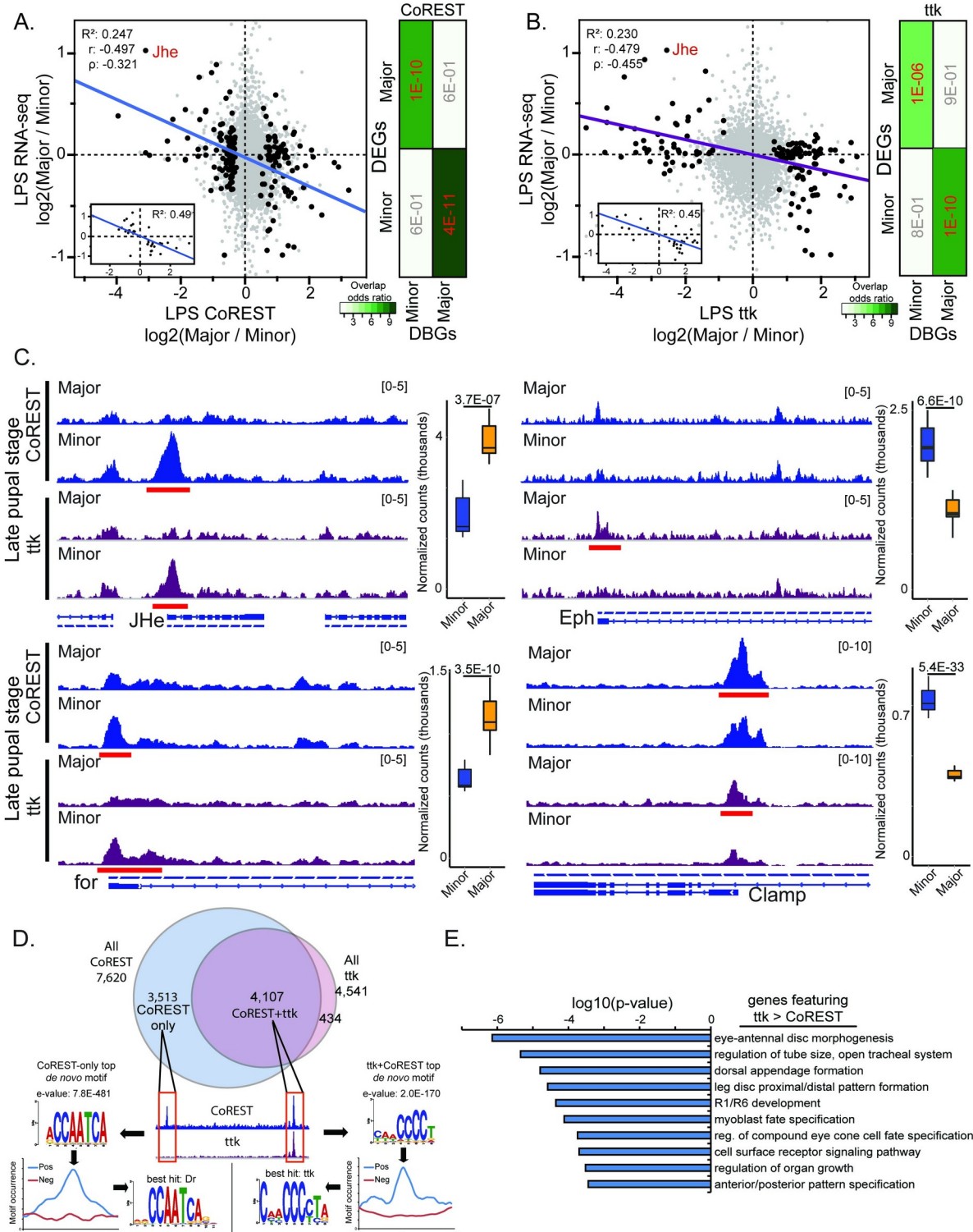

**Fig 2. CoREST and tramtrack correlate with differential expression between Major and Minor workers in late pupal stage.** A and B) correlations between log2 fold-change Major/Minor gene expression vs Major-Minor CoREST (A) and ttk (B) differences from the late pupal stage (LPS) for genes featuring a differential peak within 2kb of the TSS (black dots), as well as (inset) only those showing significant differential expression between Major and Minor. Jhe is labeled in red. Shown to the right of each plot are overlap heatmaps (odds ratios) and p-values from fisher's exact tests comparing the overlap between RNA-seq caste-biased DEGs to CoREST and ttk differentially bound genes Light gray dots denote genes with a CoREST or ttk non-differing peak to illustrate the genome-wide background. C) Example tracks of

two genes showing (left) Minor-biased binding of CoREST and ttk and Major-biased expression (Jhe top, and for bottom) as well as Major-biased binding and Minor-biased expression (top Eph, bottom Clamp). To the right of each example trackset are boxplots illustrating strength and significance of differential expression for the given gene. Tracks show reads-per-million (RPM) normalized CUT&RUN signal, and data ranges given in brackets above track pairs. D) Venn diagram of CoREST and ttk peaks and their intersection, illustrating strong co-occurrence of ttk peaks with CoREST peaks. Also shown are the top motifs associated with CoREST-only and CoREST+ttk peaks, as well as their occurrence within peaks, and best-hit known motif. E) Top 10 GO terms (Biological Process) associated with genes showing *higher* binding of ttk relative to CoREST.

in both LPS and d0, differentially expressed genes of Minor workers, but not of Major workers, showed significant enrichment for Notch-interacting genes (S1D Fig, left). Further, genes exhibiting a ttk or CoREST peak were also significantly enriched for Notch-related functions from fly (S1D Fig, right). These data suggest that not only is the relationship between ttk and Notch signaling conserved between files, but also that modulation of Notch signaling between worker castes may be one way in which ttk mediates worker division of labor.

## Pupal tramtrack shows 'poising' of d0 Minor worker repressed genes

As described above, at the LPS (Fig 2A) and at d0 stage (S2A and S2B Fig), higher CoREST/ttk co-binding in a given caste correlated with lower gene expression in the same caste. In contrast, ttk binding without CoREST in the LPS showed a striking signal predictive of later differential gene expression in d0 workers whereas CoREST did not show such a predictive pattern (Fig 3A). Indeed, ttk peak differences in the LPS between castes showed a strong negative correlation with later d0 gene expression differences (Fig 3A, right). Supporting this, when comparing genes differentially bound by ttk or CoREST in the LPS to genes differentially expressed at d0, only ttk exhibited a significant overlap with d0 DEGs (S2B Fig, left). This is contrasted by the same comparison using CoREST, where the negative correlation between LPS CoREST differential binding in LPS and d0 gene expression differences is considerably less pronounced (Figs 3A, left and S2A, left). In particular, we observed that, of regions significantly differing between pupal castes for ttk, d0 CoREST showed a striking pattern of matching this difference—but this was the case only for Minor-biased DBRs and not Major-biased DBRs (Fig 3B, right vs left). Similarly, when performing the reciprocal comparison examining pupal ttk signal at d0 CoREST DBRs, ttk showed a striking preceding pattern in LPS matching that of d0 CoREST—but as before this was the case only for Minor-biased DBRs and not Major-biased DBRs (S2C Fig, top, left vs right). This relationship is quite clear at certain individual genes showing d0 differential expression (Fig 3C). Taken together, these relationships suggest that ttk poises genes in the LPS for stable CoREST repression upon eclosion.

We found 42 genes showing Minor-biased binding of ttk (but not CoREST) in the LPS, with Minor-biased binding of CoREST only appearing at d0 (S8 Table). Strikingly, of these 42 genes, in the LPS only 5 showed Minor-repressed/Major-activated expression, while 30 genes (71%) showed later d0 Minor-repressed/Major-activated gene expression, with many showing a similar trend at d5 (Figs 3D and S2E). Relative to all differentially-bound genes, these genes showed functional enrichment for terms related to development and response to the environment (S9 Table), suggesting repression of these genes may be particularly formative to the program specifying Minor worker. These results underscore ttk pre-binding key genes in the late pupal stage to direct the later (d0) binding activity of CoREST in a caste-biased manner for subsequent gene repression at eclosion.

## Tramtrack DBRs are associated with hormonal functions

Because ttk appeared to show stronger relationships with developmental and caste-biased genes than CoREST (Fig 2E), and showed a particular relationship with predicting the Minor

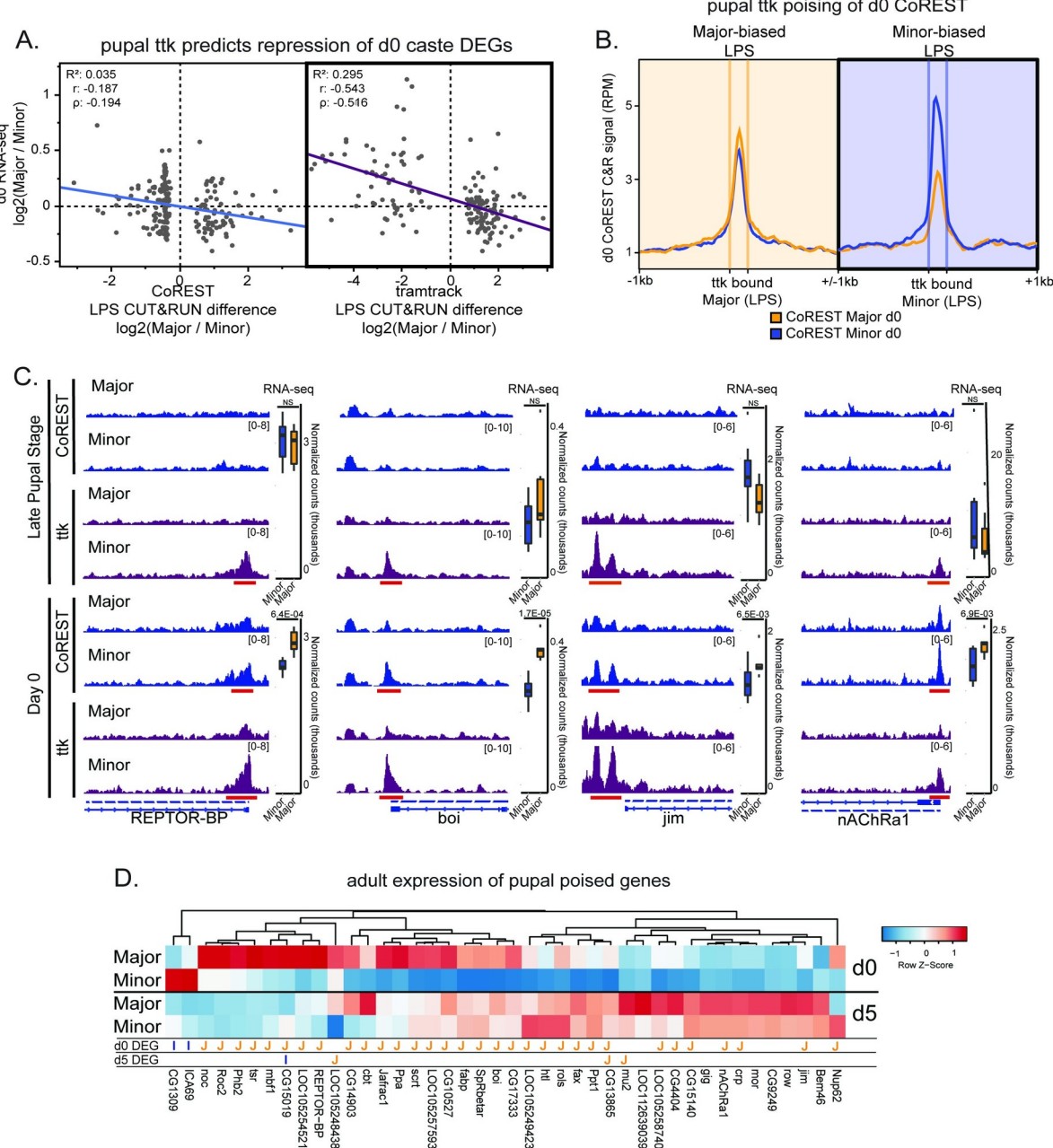

**Fig 3. Tramtrack binds a subset of developmentally-enriched CoREST peaks, and shows priming of a subset of d0 caste DEGs in pupal stage.** A) Scatterplots showing correlations between d0 caste brain differential expression (RNA-seq log2 Major/Minor) compared with CoREST (left panel) and ttk (right panel) late pupal stage (LPS) differences in binding between castes, showing that LPS ttk correlates more strongly with d0 gene expression (right) while LPS CoREST shows much a weaker association with d0 gene expression (left). B) Metaplots of CoREST d0 CUT&RUN enrichment for each caste relative to Major-biased (left plot) and Minor-biased (right plot) LPS ttk differentially bound regions, illustrating that regions preferentially bound by ttk in LPS Minors show analogous CoREST differences at d0 (right plot, bolded), while regions bound by ttk in LPS Majors do not (left plot). C) Example tracks (RPM-normalized CUT&RUN signal) of genes featuring Major-biased expression at d0, pupal differential binding of tramtrack, and d0 (but not pupal) differential binding of CoREST (red bars), suggesting ttk is poising genes in Minor pupae for repression, which are then repressed by CoREST following eclosion, and thereby Major-biased in adult gene expression. Data ranges for each track pair given in brackets. D) Heatmap of gene expression for all genes (n = 42) showing Minor-biased ttk binding in LPS but CoREST binding in d0, showing that the vast majority of such genes show Major-biased expression at either d0 or d5. Direction of DGE indicated by letter below heatmap (J = Major-biased, I = Minor-biased). Data shown represent z-score normalized average normalized count values from ≥ 6 replicates per sample type.

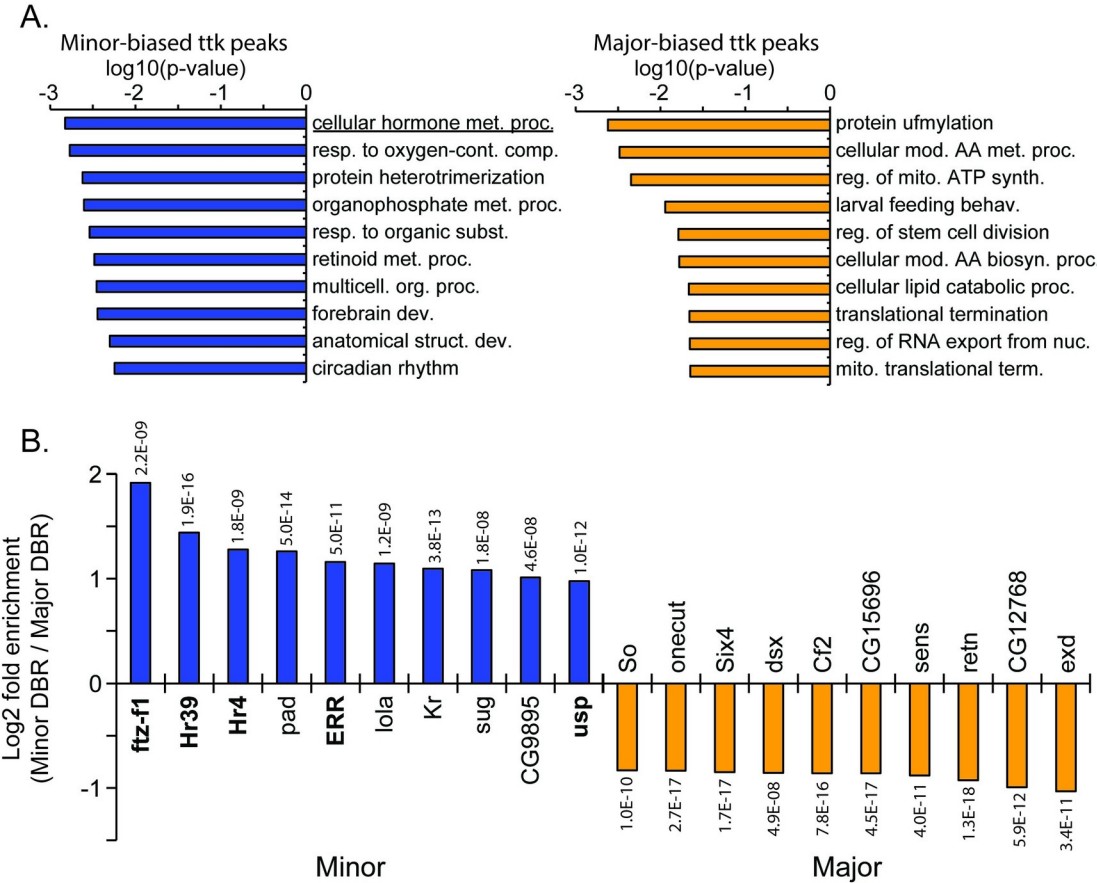

**Fig 4. Peaks differentially bound by ttk in Minor show enrichment for functions and motifs associated with hormonal signaling.** A) Top 10 GO functional terms associated with genes featuring ttk peaks showing Minor bias (left) or Major bias (right). B) Top 10 known motifs significantly enriched in Minor (left) or Major (right) ttk DBRs relative to the reciprocal caste, ranked by log2 fold difference (Minor DBR occurrence/Major DBR occurrence). Bolded motif IDs are those strongly implicated in hormonal signaling in *Drosophila melanogaster*. E-values (above/below bars) generated by AME.

worker program of d0 gene expression, we assessed functional enrichment associated with genes bound by ttk in either Major or Minor, using peaks bound at either the LPS or d0. We found that genes differentially bound by ttk in both Minor and Major workers showed functional enrichment for specific terms related to development and behavior (Fig 4A and S10 Table). In particular, we noted that the top term enriched among genes bound more highly by ttk in Minor workers was "cellular hormone metabolic process". In addition, a top differentially regulated gene here and prior [2] was a Juvenile hormone esterase. These points, and the general importance of hormones to regulation of social insect castes, led us to query ttk peaks showing significant bias to either Major or Minor for enrichment of DNA motifs, with a specific focus on DNA binding factors known to play roles in hormonal signaling. Notably we found that motifs related to transcription factors with known hormonal functions in fly were strongly over-represented among Minor-biased ttk DBRs, but not Major-biased ttk DBRs (Fig 4B). In particular Hr4 is a known downstream target of 20E signaling in fly [55] while Hr39 and ftz-f1 have been shown in fly to have target similar sites and play antagonistic roles with one another [56], and ftz-f1 has been linked to JH3 signaling in honey bee [57]. This was supported by *de novo* motif enrichment comparing Major or Minor-biased ttk peak sequences to a background of non-differing ttk peaks (S2F Fig). Taken together, these results suggest that

during this formative time in worker development, ttk differentially regulates response to hormones in Minor workers, either through repression of known hormonal response elements or association with hormone-induced co-factors.

## Discussion

Social insects of different worker castes display markedly distinct behavioral repertoires [2], in spite of a shared genome; thus caste behavioral identity is mainly dictated by non-genetic mechanisms including changes to chromatin. Despite this, chromatin-based epigenetic mechanisms directly mediating behavioral caste have been largely unknown. Previously, we demonstrated that the neuronal co-repressor CoREST mediates Major worker behavioral reprogramming towards Minor-like behavior and appeared to play a role in natural Minor worker caste differentiation, at least in part through repression of key JH3-degrading enzymes [2]. Here, we show that one binding partner of CoREST, the DNA-binding transcription factor ttk, may direct the action of CoREST to regulate a broad program of natural development of caste. By profiling gene expression as well as contrasting genome-wide binding of ttk and CoREST in the late pupal stage and just after eclosion we deduced that ttk has a central role to pre-bind to, and poise Minor-specific genes for subsequent binding of CoREST and gene repression.

We find in *C. floridanus* worker brains, that ttk and CoREST show expression bias to Minors (Fig 1C), that their binding is predictive of caste-biased repressed gene expression (Fig 2A), and that ttk binds a subset of CoREST loci (Fig 2D) enriched for development-related functions (Fig 2E). This may be similar to observations in *D. melanogaster* whereby fruitless (another BTB transcription factor) specifies sex-specific repression of genes in a cell-type specific manner, repressing genes in males that are otherwise expressed in females [58]. This is a particularly apt parallel as sex determination shares many similarities with caste specification as both specify alternative phenotypes through phenotype-specific regulation of gene expression.

While caste differences for both CoREST and ttk binding are predictive of differential gene expression (Figs 2A and 3A), ttk alone shows an interesting signature to poise caste-biased genes for repression by CoREST (Fig 3A–3D). In particular, in the late pupal stage, ttk binds a subset of genes that show subsequent repression in d0 Minors (along with d0 CoREST binding; Figs 3C, 3D and 5). Indeed, of the 42 genes bound in this manner, 32 of them show Minor-biased repression (and Major-biased expression) in d0 workers with 25 of these showing no differential expression in the late pupal stage. Further, genes bound by ttk in Minors possess hormone and development functions (Fig 4A), and relative to Major-biased DBRs, are enriched for motifs related to DNA binding factors with clear roles in hormonal signaling in *D. melanogaster* (Fig 4B). Importantly, for both CoREST and ttk only a subset of peaks significantly differed between castes (~5%), suggesting conservation of the known neurodevelopmental roles of these genes in ants, but elaboration and repurposing of ttk/CoREST function to a subset of gene targets to regulate caste-specific expression. That is, we believe that it is unlikely that caste-unbiased ttk/CoREST peaks observed reflect loci that differ between castes in a prior timepoint, and instead these likely reflect genes repressed in both castes via the conserved role of ttk/CoREST in regulating neurodevelopment and differentiation.

Our results indicate that ttk drives the function of CoREST in *C. floridanus* caste-determination, where ttk appears to direct CoREST binding to a specific subset of loci for caste-biased repression. In *D. melanogaster*, ttk has an important role in controlling cell fate [49], in particular in the nervous system [59, 60] in glial cell identity [61]. This is accomplished in part via ttk as an effector gene of Notch [60], and our observations that genes downstream of Notch

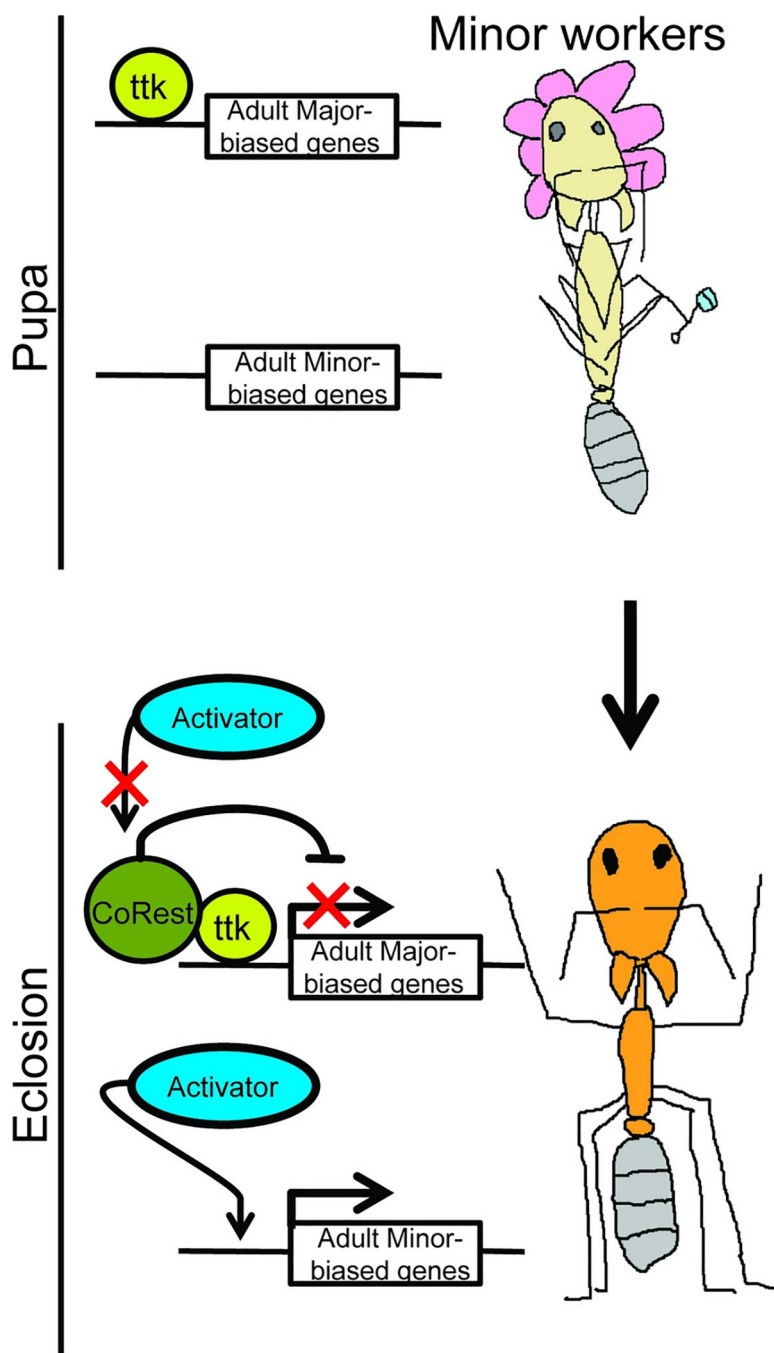

**Fig 5. Schematic of ttk-mediated recruitment of CoREST to caste-biased genes during late pupal and d0 development in Minor workers.** In late pupal stage, just prior to eclosion ttk localizes to Major-biased genes in Minor workers, where, upon eclosion, it recruits CoREST for repression of Major-biased genes.

are enriched among brain Minor-biased DEGs are consistent with these pathways (S1D Fig). We propose that Minor worker-specific repression, via this ttk-CoRest pathway, of the gene Jhe [62], an enzyme responsible for degrading the hormone JH3, may be central to mediating the behavioral differences between Major and Minor workers (Fig 2B; [2]). Our results show that CoREST/ttk-mediated repression extends beyond this single locus to target multiple

caste-biased genes throughout the *C. floridanus* genome. Furthermore, among ttk/CoREST differential peaks the enrichment of motifs associated with TFs downstream of hormonal signaling in fly highlights the important interplay between hormonal signaling and epigenetic regulation in establishing worker behavioral caste early in life. In *D. melanogaster* the relationship between ttk and Notch can be impacted by the insect hormone 20-hydroxyecdysone (20E), with 20E signaling suppressing Notch in some developmental contexts [49, 63] or acting antagonistically via ttk [52], thus it is tempting to speculate that, in *C. floridanus*, ttk activity is modulated by hormonal differences to effect or attenuate caste-specific Notch activity.

Nuclear hormone receptors typically associate with multi-component complexes comprised of other co-factors to effect transcriptional change. Given the Minor-specific enrichment of hormone signaling-related motifs (Fig 4B), CoREST/ttk may similarly collaborate with a nuclear hormone receptor and their larger multi-protein complexes. Alternatively, given the antagonistic role of 20E to JH3 [64], the predominance of 20E-related motifs (Hr4 and usp; Fig 4B) among Minor-biased differential peaks may indicate that CoREST/ttk simply acts to dampen 20E signaling that would otherwise realize a Major program of gene expression, while concomitantly increasing JH3 levels in Minor worker brains through repression of Jhe [2]. Reciprocally, shifting of the hormonal balance to 20E dominance as in Major likely engages activators associated with expression of 20E target genes. Consistent with this, the transcriptional activator cryptocephal (crc) was more highly expressed in Majors in the LPS. Importantly, in *D. melanogaster*, crc activates the 20E receptor EcR [65]; at d0 we observed more 20E-signalling related activating genes biased to Majors, including rept, Blimp-1, and MED27 (S1 Table). This is consistent with what is found in other social insects where JH3 is associated with the foraging caste while 20E is associated with the alternative caste [30, 66, 67].

Importantly, several of the TFs we observed enriched among Minor-biased ttk peaks are implicated in other systems in mediating transcriptional response to both 20E and JH3. For example, ftz-f1, is linked to JH3 signaling in multiple insects including honeybees [57, 68], but also implicated in activating genes in response to 20E [69]. Another, Ultraspiracle (Usp) is a member of the EcR heterodimer [70] but also upregulated by JH3 in honey bee [71], and may bind JHs as a homodimer [72]; Usp may, through this action, repress typical EcR targets when bound to JH3 [73]. Thus motif enrichment at Minor-biased ttk peaks alone cannot distinguish between two mechanisms: that ttk binding is either directed by these co-factors, or alternatively, that ttk is targeting these sites for repression to prevent binding by these proteins. Hence, future studies are needed to determine the factors and mechanisms mediating caste-specificity of ttk binding.

Our observations suggest that in late pupal stage, caste-specific differences in the balance between 20E and JH3 are already differently shaping the transcriptomes of Major and Minor workers to achieve caste-specific function in adulthood. Given that phenotypic differentiation between Majors and Minors begins in the larval stage it is likely that ttk is functioning downstream of a prior hormonal signal established earlier in development and acting to effect and propagate these prior caste differences during this formative neurodevelopmental stage in pupae. Future studies are needed to evaluate and understand the upstream epigenetic and hormonal signals in larvae leading to Minor-biased ttk expression, as well as to evaluate the taxonomic breadth of the ttk-CoREST mechanism in mediating social insect division of labor.

A further question is how ttk and CoREST, which maintain or repress inappropriate cell fates in differentiated cells, are leading to caste-specification in the maturing ant brain. Given the role of JH3 and 20E in modulating developmental progression, it may be that the interplay between caste-specific differences in these hormones are leading to differential development of neuronal or glial populations as Majors and Minors mature. This would be consistent with the role of ttk in controlling neuronal and glial cell fate in *D. melanogaster* [49, 59, 61, 74]. Under

this model caste-specific control of differentiation or expansion of sub-populations of neuronal or glial populations stimulated by differences in hormones may be enacted by ttk, resulting in changes in key cell populations associated with caste-specific behavior.

In conclusion our results elucidate a repressive axis that appears to govern behavioral caste identity in *C. floridanus* worker ants, and underscoring the importance of timing, hormones, and environment to realize caste-specific behavioral fate. More generally, given the strong precedent of hormones modulating behavior in many organisms, and high degree of conservation in several of the observed systems here—CoREST, and nuclear hormone receptor binding—our findings shed light on how the hormone-epigenome axis effects long-lasting behavioral plasticity across the animal kingdom.

## Methods

### Caste and age identification

Worker caste was determined as in [11]. One-day-old ants were identified by their location among brood in the nest, their general behavior, and their light cuticle coloration (compared to a reference panel of ants aged from pupation through 30 days). Late pupal stage was defined as pupa showing dark coloration within the pupal case. In order to ensure pupa were 0–1 days pre-eclosion, for each batch of RNA-seq and CUT&RUN several extra Major and Minor pupa were collected but not dissected, and checked the following day to ensure all un-dissected pupa had eclosed. In the even this was not the case (one instance) the samples from that batch were discarded. This was to ensure that ants were developmentally synchronized, as the Major workers spend more time as pupa, and thus earlier pupal timepoints risk introducing developmental desynchronization as an experimental confound. By selecting d0 and LPS we were able to choose points where Major and Minor workers are developmentally most similar in order to most consistently evaluate caste-specific differences.

### RNA isolation and preparation

For each sample type, at least two distinct colony backgrounds were used, in order to subsequently control for inter-colony variation. Individual ants were immobilized on ice for 5 minutes before brains were dissected, rinsed twice in chilled sterile Hank's balanced salt solution media (HBSS), transferred to 1.5-ml microcentrifuge tubes containing 15 μl of chilled HBSS, and immediately snap-frozen in liquid nitrogen. Total RNA was purified from individual brains by trizol extraction, followed by DNase treatment using TURBO DNase (Invitrogen). DNase-treated RNA was subsequently purified using RNase-free Agencourt AMPure XP beads (Beckman Coulter; 2:1 volume of beads:sample).

For RNA-seq, polyadenylated RNA was purified from total RNA using the NEBNext Poly (A) mRNA Magnetic Isolation Module (NEB E7490) with on-bead fragmentation as described [75]. cDNA libraries were prepared the same day using the NEBNext Ultra II Directional RNA Library Prep Kit for Illumina (NEB E7760). All samples were amplified with 8 cycles of PCR, and sequenced on a NextSeq500.

### CUT&RUN sequencing

CUT&RUN was performed on 2 Major or 3 Minor worker brains, in order to ensure similar numbers of cells for both castes. CUT&RUN was performed as in [46], but with the following modifications: Tissue was dissociated and nuclei released using buffer NE1 (20 mM HEPES-- KOH pH 7.9; 10 mM KCl; 0.5mM spermidine; 0.1% Triton X-100; 20% Glycerol) and a dounce tissue homogenizer. 300uL NE1 was added to tissue, followed by 10–15 strokes with A

then 10–15 strokes with B pestles. The Dounce and pestles was then washed with 200uL NE1, which was added to samples. Samples were centrifuged at 500g for 5min, followed by washing once with CUT&RUN was buffer (20 mM HEPES pH 7.5; 150 mM NaCl; 0.5 mM Spermidine), followed by binding to Bio-Mag Plus Concanavalin A coated beads. Antibodies were diluted 1:100 and incubated with samples overnight. In-house pA-MNase was used at a 1:200 dilution, followed by washing 2x with DIG-wash buffer (digitonin final concentration: 0.05%) and once with low salt wash buffer (20mM HEPES pH 7.5, 0.5mM spermidine, 0.05% digitonin). Cleavage was initiated by addition of 200uL of cleavage buffer (3.5mM HEPES pH7.5, 10mM CaCl2, 0.05% digitonin) and incubated in a metal heat block in ice for 5minutes, followed by brief centrifugation (100g), removal of incubation buffer, and addition of 200uL STOP buffer (170mM NaCl, 20mM EGTA, 0.05% digitonin, 50ug/mL RNase A, 25ug/mL glycogen). Samples were then purified and DNA precipitated using PCI and EtOH precipitation. Samples were resuspended in 25uL of 0.1x TE. Libraries were prepared using the NEB Ultra II DNA library prep kit with the following modifications: all reagents and volumes were half that given in the commercial protocol. For end-repair samples were incubated for 30min 20C followed by 60min at 50C. Samples were PCR amplified for 12 cycles using the recommendations of the kit with the following modification: the combined annealing/extension time was shortened to 15s for all cycles.

## RNA-seq analysis

Reads were demultiplexed using bcl2fastq2 (Illumina) with the options "—mask-short-adapter-reads 20—minimum-trimmed-read-length 20—no-lane-splitting—barcode-mismatches 0". Reads were trimmed using TRIMMOMATIC [76] with the options "ILLUMINACLIP:[adapter.fa]:2:30:10 LEADING:5 TRAILING:5 SLIDINGWINDOW:4:15 MINLEN:18", and aligned to the *C. floridanus* v7.5 assembly [77] using STAR [78]. STAR alignments were performed in two passes, with the first using the options "—outFilterType BySJout—outFilterMultimapNmax 20—alignSJoverhangMin 7—alignSJDBoverhangMin 1—outFilterMismatchNmax 999—outFilterMismatchNoverLmax 0.07—alignIntronMin 20—alignIntronMax 100000—alignMatesGapMax 250000", and the second using the options "—outFilterType BySJout—outFilterMultimapNmax 20—alignSJoverhangMin 7—alignSJDBoverhangMin 1—outFilterMismatchNmax 999—outFilterMismatchNoverLmax 0.04—alignIntronMin 20—alignIntronMax 100000—alignMatesGapMax 250000—sjdbFileChrStartEnd [SJ_files]" where "[SJ_files]" corresponds to the splice junctions produced from all first pass runs.

Differential gene expression tests were performed with DESeq2 [79]. For all pairwise comparisons the Wald negative binomial test (test ="Wald") was used for determining DEGs, using colony background as a blocking factor. Unless otherwise stated, an adjusted p-value cutoff of 0.1 was used in differentiating differentially expressed from non-differing genes in order to maximize the sensitivity of our RNA-seq results. For RNA-seq libraries at least 10M mapped reads were sequenced for each replicate. For d0, d5, and d10 RNA-seq samples, data was downloaded from the SRA (accession PRJNA530332) and analyzed as for late pupal stage RNA-seq samples.

## CUT&RUN analysis

For CUT&RUN libraries, 3 replicates were performed for each assay. Reads were demultiplexed using bcl2fastq2 (Illumina) with the options "—mask-short-adapter-reads 20—minimum-trimmed-read-length 20—no-lane-splitting—barcode-mismatches 0". Reads were trimmed using TRIMMOMATIC [76] with the options "ILLUMINACLIP:[adapter.fa]:2:30:10 LEADING:5 TRAILING:5 SLIDINGWINDOW:4:15 MINLEN:18", and aligned to the *C.*

*floridanus* v7.5 assembly[77] using bowtie2 (v2.2.6;[80]) with the option "—sensitive-local". CUT&RUN enrichment peaks were called using macs2 (v2.1.1.20160309; [81]). Differential CUT&RUN peaks were called using DiffBind [82] on the replicated samples after subtracting IgG control. DiffBind was used with the option bScaleControl = TRUE for dba.count(), bSubControl = TRUE and bFullLibrarySize = FALSE for dba.analyze(). For DiffBind, caste was used as the test category and sample batch was used as a blocking factor.

For general annotation of genes with peaks as well as motif determination (Fig 2D), peaks with fold enrichment over IgG > 2.5 were overlapped with genes. For identification of peaks/ regions featuring higher levels of ttk relative to CoREST we utilized macs2 to call peaks on ttk C&R data, using CoREST C&R data as the control library. Peaks with greater than 2.5 fold-enrichment over CoREST (n = 434) were assigned to genes. Gene ontology for these genes was tested by comparing to the set of genes featuring any CoREST peak.

### Assignment of gene orthology and functional terms

Genes (NCBI Camponotus floridanus Annotation Release 102) were assigned orthology using the reciprocal best hit method [83] to both *D. melanogaster* (r6.16) and *H. sapiens* (GRCh38) protein coding genes. Gene ontology function was assigned to genes using the blast2go tool [84] using the nr database, as well as interpro domain predictions. Gene Ontology enrichment tests were performed with the R package topGO [85], utilizing the fishers elim method, and resulting lists were reduced using ReviGO [86].

### Statistical testing

For gene overlap calculations (such as in S1D Fig) the GeneOverlap[87] R package was used. *De novo* motif discovery, enrichment calculation, and association with differentially bound regions was performed with the MEME software suite [88].

For determination of motif enrichment within DBRs, an aggregate list of peaks showing directional caste-bias in either stage were used. Any peak showing opposite differential binding between stages was discarded. AME was used to test each caste's DBRs vs the other caste's DBRs as a background, using the fly factor survey set of motif signatures [89].

All other statistical tests given in figures and tables were performed in R unless otherwise stated.

For determination of general peak motifs (Fig 2D) summits were extended by 250 bp on each side, and those with a fold enrichment > 2.5 were used for *de novo* analysis via meme-chip. Meme-chip was run against these, using a background of 10,000 randomly selected genomic windows. The top motifs were inspected for centralized enrichment, and the motif with the lowest e-value also showing central enrichment were shown in Fig 2D.

### Supporting information

**S1 Fig.** A) IP western blot of ttk using the custom antibody implemented in this study. Inputs represent 5% of IPs. B) left: similar to Fig 2D, but utilizing genes for overlaps instead of peaks, illustrating that the vast majority of genes marked with ttk are also marked with CoREST. Right: Venn diagrams showing overlap between DEGs seen between Majors and Minors at the late pupal stage (LPS) and d0 as compared to genes bound by either CoREST (top row) or tramtrack (bottom row) illustrating that genes bound by by either ttk or CoREST are significantly enriched for DEGs at both stages examined here. C) top 10 Gene Ontology (Biological Process) terms enriched among genes differentially bound (DBGs) by CoREST (left) or ttk (right) in the LPS (top row) or d0 (bottom row) as compared to genes bound by a non-differing peak in the same context. D) LPS and d0 DEGs biased to Minor workers are significantly

associated with Notch signaling in fly but Major DEGs are not (top). Likewise, genes featuring a ttk or CoREST peak are significantly enriched for fly Notch target genes (bottom). Bars represent fishers exact test odds ratio, p-values above bars are from a fisher's exact test. Results given for both all Notch targets as observed in fly, as well as those activated and repressed by Notch separately.
(TIF)

**S2 Fig.** A) Fisher's exact test odds ratios of overlap between differentially bound genes (DBGs) and differentially expressed genes (DEGs) for CoREST and ttk. Left: overlaps using DBGs in the LPS vs all DEGs presented here. Right: overlaps using d0 DBGs vs all DEGs presented here. Notably, ttk LPS binding shows far more significance of overlap with d0 DEGs as compared to the same analysis using CoREST DBGs. B) Scatterplots showing correlations between d0 caste brain differential expression (RNA-seq log2 Major/Minor) compared with CoREST (left panel) and ttk (right panel) d0 differences in binding between castes, as for Fig 3A but comparing d0 CUT&RUN differences to d0 RNA-seq differences. C) The reciprocal of Figs 3B and S2D, plotting ttk signal (RPM, replicate-averaged) for LPS (top row) and d0 (bottom row) at regions differentially bound (DBRs) by CoREST between d0 castes, illustrating that regions bound by CoREST more highly in d0 Minors shows Minor-biased enrichment of ttk in the LPS. D) Metaplots showing LPS CoREST signal for Major and Minor at regions differentially bound by ttk between LPS castes. The same regions as for Fig 3B are used but plotting LPS CoREST signal, illustrating that within-timepoint, CoREST shows the expected caste-biased enrichment at ttk caste-biased regions. E) heatmap as for Fig 3D but with late pupal stage included. F) Top three motifs derived from *de novo* motif enrichment run on either (top) Minor-biased or (bottom) Major biased ttk-peaks, compared to a background of all non-differing ttk peaks. Bolded best hits are those significantly biased in expression to the caste with the associated motif (ERR and lola).
(TIF)

**S1 Table. List of all DEGs seen between Major and Minor workers in LPS or d0 as analyzed here.** "eye_term" column indicates genes that contributed to gene ontology terms R7 cell development or eye-antennal disc morphogenesis from Figs 1A and 2E respectively.
(XLSX)

**S2 Table. Functional terms (Biological process) associated with genes significantly differing between Major and Minor workers in the LPS.**
(XLSX)

**S3 Table. All genes featuring a CoREST or ttk peak.**
(XLSX)

**S4 Table. Functional terms (BP) associated with genes given in S3 Table.**
(XLSX)

**S5 Table. Functional terms (BP) associated with genes featuring low or absent ttk binding but high CoREST binding (tested relative to a background of all genes).**
(XLSX)

**S6 Table. Functional terms (BP) associated with genes featuring low or absent CoREST binding but high ttk binding (tested relative to a background of all genes).**
(XLSX)

**S7 Table. Table showing numbers of differentially bound peaks and genes at each stage for both CoREST and ttk.**
(XLSX)

**S8 Table. List of all genes showing a significantly differing (p < 0.01) ttk or CoREST peak at d0 or LPS between Major (MAJ) or Minor (MIN).** Final column denotes genes showing poising by ttk in the LPS.
(XLSX)

**S9 Table. Functional terms (BP) associated with genes showing poising by ttk in the LPS.**
(XLSX)

**S10 Table. Functional terms (BP) associated with genes featuring a significantly Major- or Minor-biased ttk peak, denoted by the final column "Caste".**
(XLSX)

**S11 Table. DESeq2 normalized counts for all RNA-seq samples used in this study.**
(XLSX)

## Acknowledgments

We thank members of the Berger Lab for help in editing the manuscript. We thank Janko Gospocic for assistance in experimental design and general discussion.

## Author Contributions

**Conceptualization:** Karl M. Glastad, Shelley L. Berger.

**Data curation:** Karl M. Glastad.

**Formal analysis:** Karl M. Glastad.

**Funding acquisition:** Shelley L. Berger.

**Investigation:** Karl M. Glastad, Linyang Ju.

**Methodology:** Karl M. Glastad.

**Project administration:** Karl M. Glastad, Shelley L. Berger.

**Software:** Karl M. Glastad.

**Supervision:** Shelley L. Berger.

**Validation:** Karl M. Glastad.

**Writing – original draft:** Karl M. Glastad, Shelley L. Berger.

**Writing – review & editing:** Karl M. Glastad, Shelley L. Berger.

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
