## [Decision Letter · Decision Letter 0]

1 Jul 2021

Dear Dr Glastad,

Thank you very much for submitting your Research Article entitled 'tramtrack acts during late pupal development to direct ant caste identity' to PLOS Genetics.

The manuscript was fully evaluated at the editorial level and by independent peer reviewers. The reviewers appreciated the attention to an important topic but identified some concerns that we ask you address in a revised manuscript

We therefore ask you to modify the manuscript according to the review recommendations. Your revisions should address the specific points made by each reviewer.

[LINK]

Yours sincerely,

Jessica Tollkuhn

Preprint Editor

PLOS Genetics

Wendy Bickmore

Section Editor: Epigenetics

PLOS Genetics

Reviewer's Responses to Questions

**Comments to the Authors:**

Reviewer #1: In Camponotus floridanus worker ants, differences in caste behavior are programmed by epigenetic changes to gene expression driven by chromatin and transcriptional programs. For example, epigenetic factor coREST can reprogram caste behavior from soldier (major worker) to forager (minor worker). In this paper, Glastad et al. identify genome-wide transcriptional programs mediated by coREST and its binding partner ttk transcription factor complexes that differentiate minor versus major worker differences in caste behavior. The authors carry out cut and run experiments using antibodies against coREST and ttk in parallel with RNAseq from major and minor workers at late pupal stages and eclosion. They find that caste specific gene expression is apparent at pupal stages with genes involved in juvenile hormone metabolic process, and neuronal development and function as significantly different between major and minor workers. Expression of both ttk and coRESt is higher in minor caste and both genes associate with the promoters of genes associated major worker differentiation. The expression of coREST and ttk are inversely correlated with the expression of genes associated with major worker differentiation, suggesting their role in transcriptional repression of major worker genes in minor workers. Interestingly, they find that during late pupal stages, ttk is already bound at the promoters of major worker caste associated genes and coREST binding to the same sites occurs at eclosion. These suggest ttk likely primes or poises the major worker genes to be repressed in minor workers during pupal stages, and coREST binding locks in the transcriptional repressed state of major genes in minor workers to ensue transcriptional program. This is a great study unraveling the genome-wide gene regulatory programs, transcription and chromatin factors, and hormonal that contribute to differentiation of caste-specific behaviors. I think the paper is very well written and should be published in PLoS Genetics. Just a few minor comments below.

1- Fig1C. it seems like differences in jhe expression in minor versus major workers oscillates where LPS and d5 showing the biggest caste-specific difference whereas d0 and d10 shows very little. Can the authors elaborate on this with respect to the development (and critical periods) of caste specific behavior?

2- At the top of the GO terms shown in figures 1 and 2 are genes involved in R7 cell development and eye-antennal disc morphogenesis. Maybe I missed this but how do the authors think this gene cluster contributes to caste specific behaviors?

3- In Fig 3D, the transcriptional profiles of major and minor workers at day 0 is quite different.However, by d5 they look more similar between the two castes. It would be good to elaborate on this as well.

4- Like 624 “normalized signal) of genes featuring Major-biased expression at d0 but pupal differential binding of tramtrack but..”. is the first “but” supposed to be “by”?

5- A very interesting question that arises from these studies is what biases the ttk DNA binding in minor workers. Is this induced in response to pheromone detection? And what are the chromatin differences at critical genes such as jhe at LPS and day 0 minor and major workers? I do understand these might be too big as experiments to be added to this study but can be mentioned briefly in the discussion.

Reviewer #2: The review is uploaded as an attachment.

**Have all data underlying the figures and results presented in the manuscript been provided?**

Reviewer #1: Yes

Reviewer #2: Yes

PLOS authors have the option to publish the peer review history of their article (what does this mean?). If published, this will include your full peer review and any attached files.

Reviewer #1: No

Reviewer #2: No

---

## [Decision Letter · Decision Letter 1]

1 Sep 2021

Dear Dr Glastad,

We are pleased to inform you that your manuscript entitled "tramtrack acts during late pupal development to direct ant caste identity" has been editorially accepted for publication in PLOS Genetics. Congratulations!

Yours sincerely,

Jessica Tollkuhn

Preprint Editor

PLOS Genetics

Wendy Bickmore

Section Editor: Epigenetics

PLOS Genetics

Comments from the reviewers (if applicable):

Reviewer's Responses to Questions

**Comments to the Authors:**

Reviewer #1: The authors have addressed all my concerns. Very useful resource and an exciting story. Looking forward to seeing it in print.

Reviewer #2: Thanks for your thoughtful responses to our review. We are overall satisfied by the explanations and look forward to your future work.

Two notes:

-We noticed a few typos e.g. new line 184 and the headline on Figure 3A.

-In the legend for the reviewer figure showing staining of the brain with ttk antibody and anti-synapsin, we are guessing that green is ttk, as it is at least partly nuclear, and red is synapsin. However we can't understand from the legend if this is the case.

**Have all data underlying the figures and results presented in the manuscript been provided?**

Reviewer #1: Yes

Reviewer #2: Yes

PLOS authors have the option to publish the peer review history of their article (what does this mean?). If published, this will include your full peer review and any attached files.

Reviewer #1: No

Reviewer #2: No

**Data Deposition**

http://datadryad.org/submit?journalID=pgenetics&manu=PGENETICS-D-21-00586R1

**Press Queries**

---

## [Editor Report · Acceptance letter]

10 Sep 2021

PGENETICS-D-21-00586R1 

Tramtrack acts during late pupal development to direct ant caste identity 

Dear Dr Glastad, 

We are pleased to inform you that your manuscript entitled "Tramtrack acts during late pupal development to direct ant caste identity" has been formally accepted for publication in PLOS Genetics! Your manuscript is now with our production department and you will be notified of the publication date in due course.

With kind regards,

Andrea Szabo

PLOS Genetics

On behalf of:
